# The New Role of AMP-Activated Protein Kinase in Regulating Fat Metabolism and Energy Expenditure in Adipose Tissue

**DOI:** 10.3390/biom11121757

**Published:** 2021-11-24

**Authors:** Qun Wang, Jiayi Sun, Mengyu Liu, Yaqi Zhou, Lei Zhang, Yanzhang Li

**Affiliations:** 1School of Basic Medical Sciences, Henan University, Kaifeng 475004, China; wangqun011@163.com (Q.W.); Luckyzhou1206@163.com (Y.Z.); 2School of Clinical Medicine, Henan University, Kaifeng 475004, China; sunjy@henu.com.cn (J.S.); liumy@henu.edu.cn (M.L.)

**Keywords:** obesity, AMPK, brown/beige adipose tissue, energy expenditure, browning

## Abstract

Obesity is characterized by excessive accumulation of fat in the body, which is triggered by a body energy intake larger than body energy consumption. Due to complications such as cardiovascular diseases, type 2 diabetes (T2DM), obstructive pneumonia and arthritis, as well as high mortality, morbidity and economic cost, obesity has become a major health problem. The global prevalence of obesity, and its comorbidities is escalating at alarming rates, demanding the development of additional classes of therapeutics to reduce the burden of disease further. As a central energy sensor, the AMP-activated protein kinase (AMPK) has recently been elucidated to play a paramount role in fat synthesis and catabolism, especially in regulating the energy expenditure of brown/beige adipose tissue and the browning of white adipose tissue (WAT). This review discussed the role of AMPK in fat metabolism in adipose tissue, emphasizing its role in the energy expenditure of brown/beige adipose tissue and browning of WAT. A deeper understanding of the role of AMPK in regulating fat metabolism and energy expenditure can provide new insights into obesity research and treatment.

## 1. Introduction

Obesity is a disorder of nutrition and metabolism, which is triggered by genetic factors and environmental factors. Excessive accumulation of fat in the body is the main feature of obesity, which is caused by a body energy intake larger than body energy consumption. Obesity has become an increasingly severe global public health problem, due to its high incidence and the complexity of its related diseases, such as T2DM, liver steatosis, cardiovascular disease, stroke, dyslipidemia, hypertension, gallbladder problems, osteoarthritis, and certain types of cancer (endometrium, breast, ovary, prostate, liver, gallbladder, kidney, and colon) [1].

In mammalian cells, three different types of adipocytes are identified as white adipocytes, brown adipocytes, and beige adipocytes [2]. White adipocytes store excess energy in the form of triacylglycerol (TAG) and enjoy huge expansion capacity, so they are used to store energy to maintain the energy balance of the body. The excessive deposition of white adipocytes in WAT is implicated in obesity, insulin resistance, diabetes, and other metabolic diseases [3]. Contrary to white adipocytes, brown adipocytes are rich in mitochondria and are characterized by the high level of uncoupling protein 1 (UCP1) [4]. UCP1 can eliminate the membrane potential generated by mitochondria, increase oxygen consumption, and reduce ATP synthesis in the process of decoupling, and dissipate lipid to generate heat, which is an effective way to assist energy waste, thus producing a negative energy balance [5,6]. Thus, the energy expenditure will be increased by the activation of brown adipocytes. Recently, a beige adipocyte has been identified as the intermediate between BAT and WAT in mice and humans [7,8,9,10]. Theories behind the origin of beige adipocytes include (i) trans-differentiation from mature white adipocytes, (ii) existence of a distinct beige fat cell precursor, and (iii) differentiation from brown or white adipocyte precursors [5]. In the condition of chronic β-adrenergic stimuli or cold, WAT can demonstrate quite a few characteristics similar to brown adipose tissue, such as higher content of mitochondria and higher expression of UCP1, and this process is known as ‘browning’. The ‘browning’ WAT is called beige adipose tissue [11]. Beige fat is easily induced by various stimuli, such as chronic cold stimulation, exercise, and agonists of pro-adipogenic or pro-thermogenic transcription factors [12,13,14]. Considering the beneficial effects of BAT occurrence, remodeling white fat into thermogenesis-able beige fat has been considered as an attractive and promising therapeutics for the treatment of obesity and its related metabolic diseases.

AMPK plays a critical role in the regulation of adipose tissue metabolism. Current evidence confirms that AMPK activation is associated with lipogenesis/adipogenesis, fatty acid (FA) oxidation, BAT thermogenesis, and browning of WAT [15,16]. AMPK is a highly conserved and ubiquitously expressed serine/threonine protein kinase, which is a multiple heterotrimeric complexes and is made up of an α (isoforms α1 or α2) catalytic subunit, a regulatory and structurally crucial β (isoforms β1or β2) subunit, and a regulatory γ (isoforms γ1, γ2 or γ3) subunit [17]. The α (isoforms α1 or α2) catalytic subunit is comprised of an N-terminal kinase domain, an autoinhibitory domain, two regulatory-subunit-interacting domain, and a C-terminal β/γ subunits binding domain [18]. A catalytic phosphorylation site threonine-172 (Thr172) is contained on the N-terminus kinase domain of the α catalytic subunit. AMPK is activated when the Thr172 residue is phosphorylated, and this process can be induced by AMP [19] and a lot of upstream kinases, such as the calmodulin-dependent kinase kinases (CaMKKs) [20], the liver kinase B1(LKB1) [21], and transforming growth factor (TGF)-β-activated kinase-1 (TAK1) [22]. The β regulatory subunit is comprised of a glycogen-binding domain (GBD), a C-terminal scaffold domain, which promotes the combination of α and γ subunit. The β regulatory subunit is essential for maintaining the integrity of AMPK heterotrimer, and can regulate the reactivity of AMPK to the direct activators, such as A-769662 [23]. Four tandem cystathionine-beta-synthase (CBS) domains are contained in the γ subunit, and form the allosteric regulatory site of AMP/ATP on AMPK [24]. The compromised cellular energy status results in an increase in the proportion of cellular AMP: ATP, and AMP or ADP, can then bind to the CBS1 or CBS3 of the γ subunit, resulting in conformational change that activates AMPK via phosphorylation of the Thr172 site in the α subunit [25]. 5-aminoimidazole-4-carboxamide ribonucleotide (AICAR), an AMPK activator, which is an AMP mimetics, activates AMPK by interacting with γ subunit [26].

Indeed, AMPK activation is linked with a host of metabolic improvements and appears to play a vital role in mediating the beneficial effects of various pharmaceuticals/nutraceuticals [23,27,28]. In this review, in order to clarify the role of AMPK in adipose tissue, especially in regulating energy expenditure of brown/beige adipose tissue and browning of WAT, we summarize the recent studies of AMPK on fat catabolism, anabolism, and WAT browning process. Future considerations for studies examining the role of AMPK in fat metabolism and obesity are highlighted.

## 2. Mechanism of AMPK Improving Obesity

AMPK, a central regulator of cellular metabolism, mediates phosphorylation of target substrates and plays a paramount role in the regulation and maintenance of energy homeostasis, which is emerging as one of the most promising targets in the prevention and treatment of obesity based on its pivotal role in physiology and pathology [29,30]. Concerning the effect of AMPK on adipose tissue metabolism, we summarized four main mechanisms, including the role in lipogenesis/adipogenesis, lipolysis, energy expenditure, and browning of WAT.

### 2.1. The Role of AMPK in Lipogenesis/Adipogenesis

A paramount pathway of lipogenesis is the de novo synthesis (DNL), which is a complex and highly regulated metabolic pathway. In normal conditions, excess carbohydrate is converted into FA by DNL, and then the FA is then esterified into stored TAG, in the liver, and adipose tissue in the human body (Figure 1) [23]. In the cytosol, the acetyl-CoA is then converted to malonyl-CoA catalyzed by acetyl-CoA carboxylase (ACC, the first step in DNL), which is a speed limiting enzyme for FA synthesis [31]. The activity of ACC is inhibited by AMPK-mediated phosphorylation of ACC1 at Ser^79^ or ACC2 at the orthologous site Ser^212^, which reduces conversion of acetyl-CoA to malonyl-CoA and thus inhibits lipogenesis [15,32]. Therefore, AMPK activation inhibits FA synthesis by phosphorylating ACC at Ser79 to decrease the malonyl-CoA level. The ACC mutated mice have increased TAG accumulation when fed a normal chow diet compared with wide-type controls [31]. This is supported further by the observation that both fasting and exercise stimulate AMPK in rat adipose tissue with a concomitant reduction in malonyl-CoA, which may reflect inhibition of ACC [33]. After intravenous caffeine injection in rats, caffeine increased the phosphorylation of AMPK as well as ACC, and eventually increased glucose transport activity, and reduced the energy state [34]. These data indicate that removing AMPK phosphorylation of ACC leads to an increased TAG deposition through increasing FA DNL. Conversely, increasing AMPK activity or inhibiting ACC activity can improve TAG deposition by inhibiting FA synthesis (Figure 3).

In the process of lipogenesis regulated by AMPK, another target of AMPK is sterol regulatory-element-binding protein-1c (SREBP-1c), which regulates the expression of various lipogenic genes, such as, ACC1, FA synthase (FAS), and stearoyl-CoA desaturase 1 (SCD1) [35]. SREBP-1c, the primary transcriptional regulator of fat lipogenesis, is inactivated after phosphorylation at Ser372 by AMPK, which downregulates lipogenic gene expression [35].

Preadipocytes differentiate into mature adipocytes mainly through two stages: the first stage is the cessation of cell division, and the withdrawal of cell division cycle; the other stage is characterized by cell growth stagnation, the beginning of differentiation, lipid droplet appearance, and eventually differentiate into mature adipocytes [36]. The process adipogenesis is accompanied by the elevated expression of the transcription factor PPARγ, which in turn can activate C/EBPα. PPARγ and C/EBPα accelerate cell differentiation in a positive feedback manner and induce the expression of quite a few adipocyte marker genes, such as adipocyte-specific FA-binding protein and FAS [37,38].

Activated AMPK has been proposed to inhibit proliferation in several cell types in different pathways. For example, AMPK has been demonstrated to inhibit cell proliferation via inhibition of rapamycin complex (mTOR) T-cell acute lymphoblastic leukemia [39], inhibit endothelial cell proliferation via elevation of p21 and p27 expression [40], and block the growth of the HepG2 cell line via phosphorylation of p53 [41], especially the role of AMPK in preadipocyte proliferation. A multitude of studies have shown that AMPK plays a paramount role in inhibiting adipogenesis via reducing the expression of C/EBPβ (which is essential for initiation of the adipogenic transcriptional cascade), and subsequent inhibition of PPARγ, C/EBPα and late adipogenic markers such as FAS, aP2, and SREBP-1c [38,42]. Recent studies have shown that the impact of AMPK on adipogenesis may be related to the WNT/β-catenin pathway regulated by AMPK [43]. AMPK activated by AICAR increases the expression of β-catenin and nuclear accumulation, with reduced expression of adipogenic genes, such as C/EBPβ, PPARγ, C/EBPα, FAS, aP2, and SREBP-1c in 3T3-L1 adipocytes. This is reversed with siRNA-mediated knockdown of β-catenin, providing a quintessential mechanism by which AMPK inhibits adipogenesis (Figure 3) [43,44].

### 2.2. The Role of AMPK in Lipolysis

In adipocytes, the process in which TAG is hydrolyzed into glycerol and FA is called lipolysis, which takes three stages to complete. In the first place, TAG is hydrolyzed to DAG and FA by adipose triglyceride lipase (ATGL, also known as desnutrin). Second, DAG is hydrolyzed to monoacylglycerol (MAG) and FA by hormone-sensitive lipase (HSL). Third, MAG is hydrolyzed to glycerol and FA by MAG lipase (MAGL) (Figure 2) [45]. Every reaction is tightly regulated, and AMPK plays a paramount role in lipolysis.

The role of AMPK in lipolysis is controversial. Several studies report an anti-lipolytic effect of AMPK [46], whereas others suggest AMPK stimulates lipolysis [47,48]. This difference may be related to the tissue-specific function of the AMPK under different conditions. HSL is considered a speed-limiting enzyme for TAG hydrolysis, and the Ser563 and Ser660 sites of HSL can be phosphorylated to activate HSL for the lipolysis in adipose tissue, which is essential for the translocation of HSL to lipid droplets upon lipolysis stimulation. However, AMPK phosphorylated the Ser565 site of HSL and inhibited the phosphorylation at HSL Ser660 and Ser563, thereby reducing HSL activity and significantly inhibiting lipolysis in adipocytes [25]. Based on this study, it is considered that AMPK activation inhibits lipolysis (Figure 3).

However, new observations indicate that phosphorylation of HSL at Ser563 is not indispensable for the translocation of HSL to the surface of the lipid droplet [49]. While Ser565 is necessary for the translocation of HSL to the surface of the lipid droplet, because Ser565 is mutated into alanine, which leads to the inhibition of HSL translocation to the surface of the lipid droplet [49].

In addition, unused FA liberated during lipolysis returns to adipocytes to be re-esterified into TAG, which creates an energy-consuming ‘futile cycle’ [50]. In fact, it has been estimated that about 40% FA liberated during lipolysis is re-esterified into TAG in human adipose tissue [51]. FA re-esterification needs to consume a lot of ATP, which leads to an increase in the AMP/ATP ratio, while a high AMP/ATP ratio subsequently activates AMPK. Therefore, AMPK may be indirectly activated by elevation of lipolysis due to an increase in the AMP/ATP ratio [52]. This is supported by the observations that mice lacking AMPKα1 enjoy smaller adipocytes with higher basal and β-adrenergic-stimulated lipolysis rates [53]. In fact, lipolysis has been significantly inhibited by the activation of AMPK both in primary adipocytes and in vivo, as reflected by the low FA content in serum [54]. However, with the prolongation of AICAR treatment time, AMPK activation increased lipolysis. This is supported by the observations that the release of glycerol either under basal or epinephrine-stimulated conditions is potently suppressed, and the output of FA under these conditions is initially reduced and then markedly elevated as the time of exposure of adipocytes to AICAR [54]. This phenomenon can be explained by that continuous activation of AMPK promotes the expression of ATGL, which converts ATG to DAG and FA, inhibits the activation of HSL, which then converts DTG to MAG and FA [54].

Taken together, these findings suggest that AMPK has both promoting and inhibiting effects on lipolysis through different time-dependent regulation of HSL, ATGL, and other enzymes, as well as changing the expression of genes promoting lipid utilization or storage in adipocytes. The overall impact of AMPK activation on lipolysis remains controversial, and future studies are needed to elucidate the effects of chronic AMPK activation on tissue specificity and systemic lipid metabolism, which may have crucial implications for the treatment of obesity.

### 2.3. The Role of AMPK in Energy Expenditure in Adipose Tissue

Obesity is characterized by excessive accumulation of fat in the body, which is caused by a body energy intake larger than body energy consumption. Therefore, the treatment of obesity is mainly focused on reducing the energy intake and increasing the energy output [55]. However, the therapeutic strategy that reduced the energy intake demonstrated strong side effects, and perhaps instead the therapeutic approach aimed at increasing energy expenditure will be an acceptable and attractive strategy to combat obesity [55,56,57]. Three different types of adipose tissue have existed in humans, and WAT is mainly used to store excess energy, while brown and beige adipose tissue is related to energy expenditure. Therefore, the strategy of increasing the thermogenic capacity of beige adipose tissue and brown adipose tissue may be the most effective and attractive treatment for obesity in humans [58,59].

As in a multitude of obese patients, AMPK activity is significantly impaired in a variety of obese animal models [60,61,62]. When stimulated by cold or β-adrenaline, the capability of adaptive thermogenesis and energy expenditure have been significantly attenuated in mice with adipose tissues AMPKα knockout [14]. The cold-induced thermogenic genes expression has been dramatically impaired in adipose tissues lacking AMPKα [14]. A study demonstrated that AMPK β1β2 adipose tissue-specific null (AMPKβ1β2-AKO) mice were generated and were found to develop an increase in high-fat diet (HFD)-induced insulin resistance and hepatic steatosis due to compromised BAT and WAT function [14,63]. Another study reported that adipose tissue-specific deletion of both AMPK α1 and α2 subunits (AMPKα1α2-AKO) are found to reduce lipolysis under basal conditions and increase lipolysis in adipose tissue treated with isoproterenol [15,64]. This indicates that AMPKα is required for adipose tissue thermogenesis and energy expenditure. Moreover, inguinal white adipocyte is promoted by adopting metabolic characteristics similar to brown adipocyte by AMPK activated by A-769662, which protects from HFD-induced obesity [14]. Collectively, these findings indicate that the inhibition of AMPK may lead to obesity and other related metabolic diseases, while activated AMPK may improve them.

Adipose tissue thermogenesis and energy expenditure are regulated by the expression of thermogenic genes, such as UCP-1, peroxisome proliferator-activated receptor gamma coactivator 1 alpha (PGC1α), which promote the development of brown and beige adipocyte [65]. Brown adipose tissue (BAT) consumes significant amounts of chemical energy through uncoupled respiration and thermogenesis mediated by the major thermogenic factor UCP-1 [66], which raises the interest in stimulating thermogenesis therapeutically for the treatment of metabolic diseases associated with obesity. PGC1α, another factor that regulates adipose tissue thermogenesis and energy expenditure, promotes the development of brown and beige adipocytes [65]. It is proved that the diabetes medication canagliflozin induces mitochondrial biogenesis and function through the AMPK-Sirt1-Pgc-1α signaling pathway, which directly increases cellular energy expenditure of adipocytes [67]. The role of AMPK alluded in brown adipogenesis is mainly focused on differentiation. CIDEA is involved in the degradation of the AMPKβ subunit through regulating the proteasome activity dependent on ubiquitin. When CIDEA expression is knocked out in mice, AMPK expression and activity are increased in BAT, and the mice accordingly had increased energy expenditure, and obesity triggered by an HFD was suppressed [68]. In addition, it has been recently confirmed that AMPK increased the development of brown adipocytes with a high abundance of UCP-1 [69]. Consistent with the phenotype observed when AMPKα activity is inhibited, brown adipocyte progenitor cells are profoundly decreased and brown adipogenesis is inhibited in AMPK deficient mice [70,71]. In general, there is growing evidence on the important role of AMPK as a promoter of the development of brown adipocytes.

Furthermore, UCP-2 and UCP-3 are also involved in the regulation of energy metabolism as UCP homologs. In animals with an HFD, exercise, and hibernation, the mRNA expression levels of UCP-2 and UCP-3 increased accordingly to promote glucose and fatty acid metabolism. The data from human genetic studies and measurements support the contribution of UCP2 and UCP3 to resting energy expenditure. Therefore, enhancing the expression of UCPs may increase energy expenditure and provide a defensive effect on disordered lipid metabolism and damage [72,73].The mammalian target of mTOR, an essential regulatory factor of cell growth, proliferation, and metabolism, the same as AMPK, plays a critical role in the process of preadipocyte differentiation [74]. AMPK activity is dependent on low-energy conditions, such as low ATP/AMP ratio, while mTOR is activated by diverse growth-positive signals, such as high ATP/AMP ratio [17,75]. mTOR promotes DNL through sterol responsive element binding protein (SREBP) transcription factors, promoting growth by promoting the transition of glucose metabolism from oxidative phosphorylation to glycolysis, and obese or HFD treated mice have more elevated mTOR signaling in many tissues, including the pancreas [76]. In fact, the mTOR activity is inhibited by AMPK through the phosphorylation of tuberous sclerosis complex 2 (TSC2), which is a critical regulatory protein of mTOR activity [77,78]. An early study demonstrates that the inhibition of mTOR by AMPK is essential for the differentiation of brown adipocytes in the early stage of differentiation [78]. In addition, it has been observed that brown adipose tissue and energy expenditure are increased, and the resistance to obesity induced by a high-fat diet is enhanced in mice with Raptor knockout in adipocytes [79]. However, persistent mTOR inhibition weakens the function of brown adipose tissue and increases fat accumulation [80,81,82]. These reveal a cross regulating role of the AMPK and mTOR pathway in the differentiation and function of brown adipocytes (Figure 3).

Differentiation of brown adipose tissue regulated by AMPK may be related to DNA methylation and hydroxymethylation, which are closely associated with the cellular metabolic status and enzyme cofactors [83]. It is observed that brown adipose tissue mass, UCP-1 expression, and mitochondrial content are all lower in neonatal mice with AMPKα1 ablation. Interestingly, high levels of DNA methylation and low levels of hydroxymethylation are also observed in the Prdm16 promoter in these AMPKα1 deficient neonatal mice [71]. AMPKα1 ablation reduced isocitrate dehydrogenase 2 activity and cellular α-ketoglutarate levels, a key metabolite regulator for DNA demethylation mediated by ten-eleven translocation hydroxylases. Therefore, the differentiation of brown adipose tissue is related to the methylation of the Prdm16 promoter, which is associated with the increase of IDH2 activity induced by AMPK (Figure 3) [71].

The role of AMPK in adipose tissue can be further confirmed by the complete ablation of the AMPK alpha subunit or beta subunit in cultured adipocytes or mice adipose tissue. Since 2016, mice with the complete ablation of AMPK alpha1 and alpha2 subunits, or AMPK beta1 and beta2 subunits in adipose tissues, have been produced [14,63]. Mice with the complete ablation of AMPK alpha1 and alpha2 subunits (AMPKα AKO mice) were cold intolerant, and their inguinal WAT displayed impaired mitochondrial integrity and biogenesis, decreased energy expenditure and oxygen consumption, and reduced expression of thermogenesis-related genes upon cold exposure, such as UCP-1, CIDEA, Cox8b, Cox7a1, and Ppargc1a [14]. Mice with complete ablation of AMPK beta1 and beta2 subunits (AMPKβ AKO mice) were cold intolerant and resistant to β-adrenergic activation of BAT and browning of WAT [63]. These results demonstrate that AMPK plays a vital role in the browning process in inguinal WAT and regulates whole-body energy homeostasis, which suggests that the directly targeted activation of adipocyte AMPK is likely to be therapeutically viable. Further research reveals that the number of mitochondrion has been significantly affected by the ablation of adipocyte AMPK. The specific ablation of AMPK beta1 and beta2 subunits in adipose tissue, does not alter the mitochondrial number of BAT, but reduces the mitochondrial quality, which is characterized by large swollen mitochondrial and disrupted cristae in morphology [63]. Thus, the impact of the adipose tissue-specific ablation of AMPK on energy expenditure is implicated in the regulation of mitochondrial function by AMPK.

### 2.4. The Role of AMPK in Browning of WAT

Under the stimulation of certain factors, white adipocytes will be transformed into beige adipocytes, which is called ‘browning’. When these beige adipocytes are not activated, they exhibit properties similar to white adipocytes. Together with white adipocytes, they serve as an energy reservoir to release energy in the form of free FA and play an important role in glucose and other metabolisms. Once activated, beige adipocytes will acquire many dense mitochondria with high UCP1 expression and high thermogenic capacity similar to brown adipocytes for core thermoregulation, etc. [84].

In recent years, due to beige adipocytes having a heat-generating ability similar to brown adipocytes, there has been an increasing interest in the browning of fat. Although exercise, cold exposure, or beta-adrenergic have been considered as the most common inducers for fat browning, there are other inducible factors, such as, beta amino isobutyric acid, inflammatory stress, gamma amino butyric acid, hypoxia, PPARɣ agonists, JAK inhibition, and irisin (a cleavage product of Fndc5 gene) [8,85,86,87,88]. AMPK is a key regulator of brite cell formation in browning. Aliki et al. performed lentivirus-mediated short hairpin knockout and experiments with pharmacological inhibitors to prove that AMPK promotes the formation of UCP1-rich brown/beige adipocytes [69]. The change or disappearance of any subunit of AMPK will affect the appearance of brown/beige adipocytes, as shown through knock-down specific subunits. AMPKβ AKO mice enjoy an impaired browning ability to decrease mitochondrial content in the WAT, so only a small amount of beige fat can be produced [23]. Wu et al. found that a lack of adipocyte AMPKα induced thermogenesis and obesity under cold and over-nutrient conditions [14]. Consistent with these observations, numerous other studies have found that indirectly increasing AMPK activity in WAT also results in the browning of WAT and increases in energy expenditure, such as, folliculin, berberine, xanthohumol, myostatin, and liver kinase B1 [89,90,91,92,93]. In addition, the AMPK in the ventromedial nucleus of the hypothalamus (VMH) regulates thermogenesis by manipulating the sympathetic nerve firing to beige adipose tissue [94]. Phytol administration stimulates the browning of mice inguinal subcutaneous WAT, with an increased expression of brown adipocyte marker genes (UCP1, PRDM16, PGC1α, PDH, and Cyto C), which activated the AMPKα signaling pathway in mice inguinal subcutaneous WAT and 3T3-L1 cells [95,96]. Consistent with the result, the inhibition of AMPKα with Compound C (dorsomorphin, a selective AMPK inhibitor) abolished phytol-stimulated brown adipogenic differentiation and the formation of brown-like adipocytes [95,97]. These demonstrate that AMPK plays an important role in the browning of WAT. However, whether AMPK activation is a necessary and sufficient condition for browning phenotypes of white fat and the specific mechanism of AMPK activation inducing browning, are still not entirely clear. Future research should focus on improving the browning level of WAT and the particular mechanism of AMPK promoting the browning of WAT.

## 3. Other Functions of AMPK

In addition, besides the role in lipid metabolism, AMPK also regulates a variety of physiological metabolic processes, such as carbohydrate metabolism, protein metabolism, cell polarity, growth, apoptosis, and ferroptosis [98,99,100,101]. Correspondingly, the AMPK has been associated with a wide range of pathological conditions, such as, aging and longevity, obesity and metabolic syndrome, cardiovascular disease and reperfusion injury, cancer, dementia, neurogenesis, and stroke [98,102]. For aging and longevity, AMPK regulates the role of some downstream nutritional sensors, and ultimately controls the cell and physiological processes, such as, mTOR, the insulin/insulin-like growth factor signaling pathway (IIS), and the sirtuins, which directly or indirectly participates in the regulation of some signal, including AMPK, PGC1α, and mTOR, to delay cellular senescence by deacetylating some key proteins [103,104]. For cardiovascular disease and reperfusion injury, the activation of the AMPK signaling pathway is a protective mechanism in the ischemia/reperfusion injury, which might be associated with the reduction of protein synthesis triggered by the inactivation of eIF2α [105,106]. In addition, the activated AMPK elevates the level of glucose-regulated protein 78, reduces the expression of pro-apoptotic molecules CHOP and caspase 3, increases the ratio of Bcl2/Bax, and inhibit the ERS-induced apoptosis [107]. Activated AMPK could also inhibit cardiomyocyte apoptosis [108]. For cancer, AMPK is closely correlated to the tumor-suppressive functions of LKB1 and P53, consequently modulating the activity of cell survival signaling such as mTOR and Akt, leading to cell growth inhibition and cell cycle arrest [109,110,111]. Dementia, neurogenesis, and stroke are neurodegenerative disorders characterized by a progressive degeneration of nerve cells, eventually leading to disease. It has been reported that neurodegenerative disorders might be correlated with the overactivated AMPK [112,113,114]. At the systemic level, AMPK helps control appetite, energy expenditure, and substrate utilization in response to exercise, nutrients, and cytokines may be linked to an ocean of physiological processes. AMPK signaling may be altered in a number of disease conditions; however, despite the convincing associations between AMPK signaling under a number of different conditions and treatments, there is an urgent need for studies on the exact mechanism of AMPK.

## 4. Advances in AMPK Targeted Drugs

More and more AMPK targeted drugs are currently appearing in front of the public, but due to various challenges, such as the complexity of different combinations of AMPK subunit isoforms in tissues and cells of different species, the phosphorylation events of AMPK activation and the safety of AMPK activation are still unclear [115], as currently only one has been approved by the regulatory authorities to enter the market and more are in clinical trials. Metformin, launched in 1959, has become an excellent first-line pharmacologic treatment for type 2 diabetes in most patients. Unlike most modern drugs, metformin is derived from natural products used in herbal medicine, rather than designed for specific pathways or mechanisms [116]. One of its prominent therapeutic effects is the inhibition of liver gluconeogenesis. The most widely studied mechanism of the drug is the activation of the signal kinase AMPK, so as to achieve the main therapeutic effect of inhibiting hepatic gluconeogenesis. In addition, studies have shown that metformin exerts its anti-obesity effects by increasing mitochondrial biogenesis, reducing fatty acid intake, and stimulating brown fat metabolism to produce heat [117]. Therefore, the fixed-dose combination of metformin and sibutramine has undergone a phase III clinical study in Silanes to treat obesity. However, because sibutramine may increase the risk of heart disease and stroke among users, the Committee for Medicinal Products for Human Use (CHMP) of the European Medicines Agency suspended the sale and use of all weight-reducing drugs containing sibutramine in the European Union in 2010, so there have been few reports on this study recently. Fluoxetine, a drug under preclinical study and mainly used to treat various depression, has been proved to have remarkable antiproliferative activity and induce autophagy death in breast cancer cells. The mechanism of its autophagy death is related to the inhibition of eEF2K and the activation of the AMPK-mTOR-ULK complex axis [118]. Metformin combined with fluoxetine can enhance the antidepressant effect of fluoxetine, and its mechanism may be related to the activation of AMPK and cyclic adenosine monophosphate response element binding protein (CREB) [119]. Although many AMPK-related drugs have not yet been marketed due to various challenges, they still have huge therapeutic potential. Finally, the summary of other AMPK-related drugs is shown in Table 1.

## 5. Conclusions

AMPK is present in all tissues as αβγ heterotrimer, which is regulated by multiple genes encoding each of the subunits (α1, α2, β1, β2, γ1, γ2, γ3) with differential tissue-specific expression and activity. AMPK regulates cellular energy metabolism through direct effects on gene transcription and key metabolic enzymes. There is now clear and compelling evidence that AMPK plays an important role in regulating adipose tissue metabolism, especially in regulating the energy expenditure of brown adipose tissue and beige adipose tissue.

However, there are still quite a few important issues that require further study. Firstly, AMPK integrates a multitude of inputs and acts on an ocean of different substrates, and is a central energy sensor in vivo, which is involved in the regulation of multiple metabolic pathways. Determining the predominant effectors of AMPK mediated browning of WAT and non-shivering thermogenesis will be an issue for future studies. Secondly, whether beige adipocytes can contribute to whole-body energy expenditure significantly in vivo remains controversial, particularly as accumulating evidence exists that the thermogenic contribution might be insignificant [88,120]. Furthermore, as studies have shown that AMPK exerts opposite effects on the central nervous system and peripheral metabolism, activation of the system is not the best strategy to treat obesity, so site-specific operations on AMPK are necessary. In addition, beige fat might enjoy a different physiological function than thermogenesis. This is supported by the fact that it can be induced by exercise [121]. The role of AMPK in the browning of adipocytes remains to be studied further.

## Figures and Tables

**Figure 1 biomolecules-11-01757-f001:**
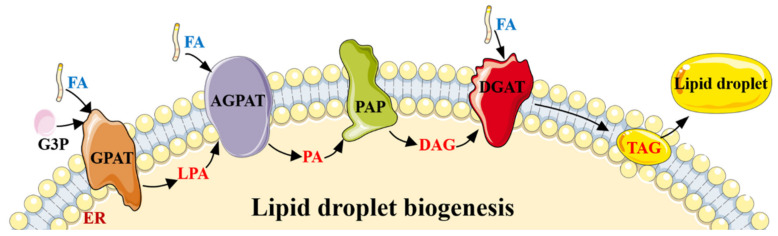
The lipogenesis process by DNL. Lipid droplet biogenesis process. Glycerol-3-phosphate (G3P) and one molecule of activated FA generate lysophosphatidic acid (LPA), under the action of glycerol-3-phosphate acyltransferase (GPAT). LPA and another molecule of activated FA generate phosphatidic acid (PA) catalyzed by 1-acylglycerol-3-phosphate O-acyltransferase (AGPAT), and then PA is converted into diacylglycerol (DAG) by phosphatidate phosphatase (PAP). Then, diacylglycerol acyltransferase (DGAT) converts DAG and one molecule of activated FA into TAG. The whole process occurs in the endoplasmic reticulum (ER), and the generated TAG is released between the lipid bilayers of the ER membrane. When TAG is accumulated to a certain extent, new lipid droplets sprout from the ER and are released into the cytoplasm.

**Figure 2 biomolecules-11-01757-f002:**
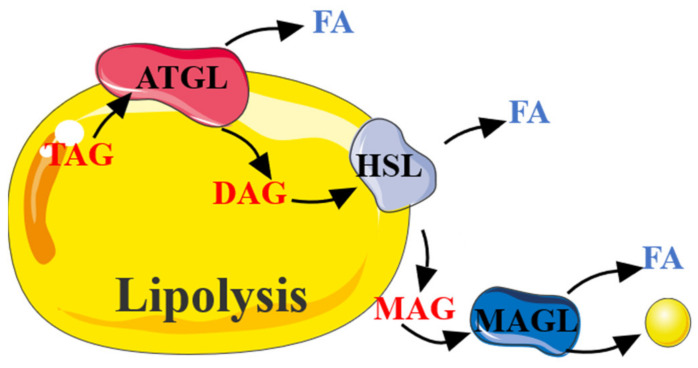
The process of lipolysis. TAG was decomposed by continuous catalysis of ATGL, HSL, and MAGL to release FA.

**Figure 3 biomolecules-11-01757-f003:**
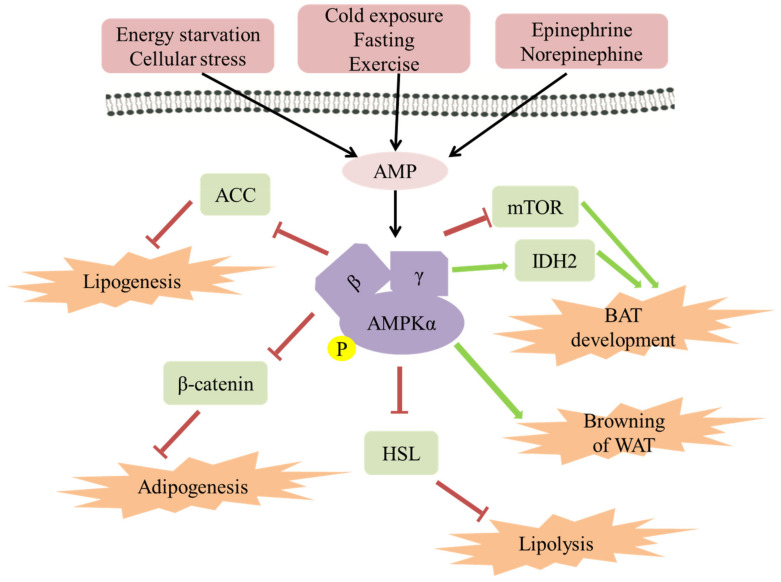
The role of AMPK in regulating fat metabolism and energy expenditure in adipose tissue. AMPK has been activated by AMP in a starvation or stress state. It has been established that AMPK can be activated by cold exposure, fasting, and exercise, in a β-adrenaline-dependent manner. Activated AMPK leads to ATP production and a series of metabolic changes, for example, lipogenesis, adipogenesis, lipolysis, BAT energy expenditure, and browning of WAT.

**Table 1 biomolecules-11-01757-t001:** AMPK targeted drugs in obesity therapy.

Drug Name	Highest Phase	Molecular Mechanism	Therapeutic Group
Metformin hydrochloride	Launched-1959	AMPK Activators, Calcium Channels alpha2/delta Subunit Ligands, PDE5A Inhibitors, Insulin Sensitizers	Anti-obesity Drugs, Type 2 Diabetes
Sibutramine/Metformin	Phase III	5-HT Reuptake Inhibitors, AMPK Activators	Anti-obesity Drugs
L-leucine/sildenafil citrate	Phase II	AMPK Activators, Calcium Channels alpha2/delta Subunit Ligands, PDE5A Inhibitors, Insulin Sensitizers	Anti-obesity Drugs
Metformin hydrochloride/sildenafil citrate/L-leucine	Phase II	AMPK Activators, Calcium Channels alpha2/delta Subunit Ligands, PDE5A Inhibitors, Insulin Sensitizers	Anti-obesity Drugs, Liver and Biliary Tract Disorders
R-17	Preclinical	AMPK Activators	Anti-obesity Drugs, Liver and Biliary Tract Disorders
BC-1618	Preclinical	AMPK Activators, FBXO48 Inhibitors, Insulin Sensitizers	Anti-obesity Drugs
MK-3903	Preclinical	AMPK Activators, Insulin Sensitizers	Anti-obesity Drugs, Type 2 Diabetes
Fluoxetine hydrochloride/metformin hydrochloride	Preclinical	AMPK Activators, PDE5A Inhibitors, CYP3A4 Inhibitors, Insulin Sensitizers, SERT Inhibitors, Voltage-Gated Sodium Channel Blockers	Anti-obesity Drugs
C-455	Preclinical	AMPK Activators	Anti-obesity Drugs, Antidiabetic Drugs, Cardiovascular Diseases, Lipoprotein Disorders, Liver and Biliary Tract Disorders
di-Metformin glutamate docosahexaenoate	Preclinical	AMPK Activators, Insulin Sensitizers	Anti-obesity Drugs, Type 2 Diabetes
R-419	Preclinical	AMPK Activators, Insulin Sensitizers, Complex I Inhibitors	Anti-obesity Drugs, Non-Opioid Analgesics, Oncolytic Drugs, Type 2 Diabetes
pCMV-AdipoR2	Preclinical	AMPK Activators	Antidiabetic Drugs, Anti-obesity Drugs
Baccharin	Preclinical	AKR1C3; 17beta-HSD5 Inhibitors, AMPK Activators	Anti-obesity Drugs, Metabolic Disorders, Type 2 Diabetes
Ampkinone	Preclinical	AMPK Activators	Antidiabetic Drugs, Anti-obesity Drugs
DNP-60502	Preclinical	AMPK Activators	Antidiabetic Drugs, Anti-obesity Drugs
Panduratin A	Preclinical	AMPK Activators, NFKB Activation Inhibitors	Antiarthritic Drugs, Antibacterial Drugs, Anti-obesity Drugs, Atopic Dermatitis, Disorders Oncolytic Drugs
cis-3’,4’-Diisovalerylkhellactone	Preclinical	AMPK Activators, GAA Inhibitors, NO Production Inhibitors, PLA2 Inhibitors, PAFR Antagonists	Antidiabetic Drugs, Anti-obesity Drugs, Antiplatelet Therapy, Inflammation

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
