# Peer review of "The New Role of AMP-Activated Protein Kinase in Regulating Fat Metabolism and Energy Expenditure in Adipose Tissue"

_biomolecules, 2021, doi:10.3390/biom11121757_

Round 1

Reviewer 1 Report

In the manuscript entitled: “The new role of AMP-activated protein kinase in regulating fat metabolism and energy expenditure in adipose tissue” Wang et al review the role of AMPK in detail in lipogenesis/adipogenesis, lipolysis, energy expenditure in adipose tissue, and browning of WAT. They also discussed other biofunctions of AMPK, recent advances in AMPK targeted drugs, and future perspectives about AMPK research. Since obesity has become an increasingly serious global public health problem, the discussion and evaluation of AMPK related regulation of metabolism and energy expenditure in adipose tissue is of need and is certainly of interest for the obesity therapy field and merits publication.

Minor Points:

  1. In the paragraph just below Figure 2, the authors claimed HSL is considered to be a speed-limiting enzyme for TG hydrolysis. Are there any references that support this?
  2. In the second paragraph of part 5, “Consistent with the result, the inhibition of AMPKα with Compound C totally abolished phytol-stimulated …”. What’s Compound C?
  3. In part 7, “More and more AMPK targeted drugs appear in front of the public, but due to various challenges, currently only one has been approved ….” Could the authors briefly describe some existing detailed challenges?

Author Response

Dear reviewer,

On behalf of my co-authors, we thank you very much for your positive and constructive comments and suggestions on our manuscript entitled ‘The new role of AMP-activated protein kinase in regulating fat metabolism and energy expenditure in adipose tissue’. (ID: biomolecules-1449991).

We have studied the comments carefully and have completed the revision and marked it in the revision mode. We have tried our best to revise our manuscript according to the comments, and our reply to the comments is as follows:

  1. In the paragraph just below Figure 2, the authors claimed HSL is considered to be a speed-limiting enzyme for TG hydrolysis. Are there any references that support this?

Response: Thank you for your valuable and thoughtful comments. There have been many studies support that HSL is considered to be a speed-limiting enzyme for TG hydrolysis as follows:

â‘ Adipose triglyceride lipase (ATGL) and hormone-sensitive lipase (HSL) are rate-limiting enzymes that control the hydrolysis of TG [1].

â‘¡In both in vitro models, BSCFAs and SCFAs reduced phosphorylation of hormone sensitive lipase, a rate limiting enzyme in lipolysis [2].

â‘¢Hormone-sensitive lipase (HSL) is rate limiting for diacylglycerol and cholesteryl ester hydrolysis in adipose tissue and essential for complete hormone-stimulated lipolysis [3].

[1] D. Jiang, D. Wang, X. Zhuang, Z. Wang, Y. Ni, S. Chen, and F. Sun, Berberine increases adipose triglyceride lipase in 3T3-L1 adipocytes through the AMPK pathway. Lipids Health Dis 15 (2016) 214.

[2] E. Heimann, M. Nyman, A.K. Palbrink, K. Lindkvist-Petersson, and E. Degerman, Branched short-chain fatty acids modulate glucose and lipid metabolism in primary adipocytes. Adipocyte 5 (2016) 359-368.

[3] W.J. Shen, Z. Yu, S. Patel, D. Jue, L.F. Liu, and F.B. Kraemer, Hormone-sensitive lipase modulates adipose metabolism through PPARgamma. Biochim Biophys Acta 1811 (2011) 9-16.

  1. In the second paragraph of part 5, “Consistent with the result, the inhibition of AMPKα with Compound C totally abolished phytol-stimulated …”. What’s Compound C?

Response: Thank you for your valuable and thoughtful comments. We are very sorry for our negligence that there is no comprehensive elaboration in our essay. Compound C is dorsomorphin, which has been widely used in cell-based, biochemical, and in vivo assays as a selective AMPK inhibitor. We have modified it as follows:

‘Consistent with the result, the inhibition of AMPKα with Compound C (dorsomorphin, a selective AMPK inhibitor) totally abolished phytol-stimulated brown adipogenic differentiation and formation of brown-like adipocytes.’

The change is on page 9.

  1. In part 7, “More and more AMPK targeted drugs appear in front of the public, but due to various challenges, currently only one has been approved ….” Could the authors briefly describe some existing detailed challenges?

Response: Thank you for your valuable and thoughtful comments. Considering the Reviewer’s suggestion, we have briefly described some of the existing detailed challenges as follows:

‘More and more AMPK targeted drugs appear in front of the public, but due to various challenges, such as, the complexity of different combinations of AMPK subunit isoforms in tissues and cells of different species, the phosphorylation events of AMPK activation are still unclear, the safety of AMPK activation and so on [115], currently only one has been approved by the regulatory authorities to enter the market and more are in clinical trials.’

[115] S. Olivier, M. Foretz, and B. Viollet, Promise and challenges for direct small molecule AMPK activators. Biochem Pharmacol 153 (2018) 147-158.

The change is on page 9.

Please see the attachment about the revised manuscript.

We would like to express our great appreciation for your comments on our paper. 

Thank you and best regards.

Yours sincerely,

Lei Zhang, Ph.D.

Reviewer 2 Report

The review submitted by Wang Qun et al. addresses the emerging role of AMPK in adipose tissue through the regulation of fat metabolism and energy expenditure; the Authors have nicely covered the growing literature in this area. Clarifying the mechanisms underlying BAT activation and the "browning" process is of great interest for the development of strategies against obesity. In addition, the report provides a comprehensive overview of all the literature in the sector, addressing recent developments in a pertinent way.

Sirtuins 1 and 3 play a crucial role in the regulation of thermogenesis by modulating multiple aspects of cell metabolism, mitochondrial biogenesis and homeostasis. The referee believes that the review might be improved by adding one comment on the axis AMPK/PGC1α/Sirt1/3

The authors could mention the involvement of UCP3 in the regulation of energy metabolism and weight control. Further investigations into the molecular role of UCP3 in BAT are relevant for the future and could provide new insights into its function.

This referee suggest to add a figure in paragraph 5 about the role of AMPK in browning of WAT.

The referee has found many typing errors (marked in yellow in the attached file).

The number of paragraphs is not correct.

Other minor points:

Introduction, 12 lines TAG not TAH

Legend figure 1: add the full name for GPAT and PAP. Add diacylglycerol after the first DAG, not after the second.

page 5, 5 lines from bottom; please replace to enjoy with to develope

page 8, 15 lines: irisin not IRISIN

page 8, 27 lines: add Ventromedial Nucleus of the Hypothalamus (VHM)

Author Response

Dear reviewer,

On behalf of my co-authors, we thank you very much for your positive and constructive comments and suggestions on our manuscript entitled ‘The new role of AMP-activated protein kinase in regulating fat metabolism and energy expenditure in adipose tissue’. (ID: biomolecules-1449991).

We have studied the comments carefully and have completed the revision and marked it in the revision mode. We have tried our best to revise our manuscript according to the comments, and our reply to the comments is as follows:

  1. Sirtuins 1 and 3 play a crucial role in the regulation of thermogenesis by modulating multiple aspects of cell metabolism, mitochondrial biogenesis and homeostasis. The referee believes that the review might be improved by adding one comment on the axis AMPK/PGC1α/Sirt1/3

Response: Thank you for your valuable and thoughtful comments. As the Reviewer’s suggestion, sirtuins are related to multiple thermogenic regulation, so we added some comment on the axis AMPK/PGC1α/Sirt1/3, the changes are as follows:

‘It is proved that the diabetes medication canagliflozin induces mitochondrial biogenesis and function through AMPK-Sirt1-Pgc-1α signaling pathway, which directly increases cellular energy expenditure of adipocytes [67].’

‘the sirtuins, which directly or indirectly participates in the regulation of some signal, including AMPK, PGC1α and mTOR, to delay cellular senescence by deacetylating some key proteins [103].’

[67] X. Yang, Q. Liu, Y. Li, Q. Tang, T. Wu, L. Chen, S. Pu, Y. Zhao, G. Zhang, C. Huang, J. Zhang, Z. Zhang, Y. Huang, M. Zou, X. Shi, W. Jiang, R. Wang, and J. He, The diabetes medication canagliflozin promotes mitochondrial remodelling of adipocyte via the AMPK-Sirt1-Pgc-1alpha signalling pathway. Adipocyte 9 (2020) 484-494.

[103] C. Chen, M. Zhou, Y. Ge, and X. Wang, SIRT1 and aging related signaling pathways. Mech Ageing Dev 187 (2020) 111215.

The changes are on page 6 and page 9 respectively.

2.The authors could mention the involvement of UCP3 in the regulation of energy metabolism and weight control. Further investigations into the molecular role of UCP3 in BAT are relevant for the future and could provide new insights into its function.

Response: Thank you for your valuable and thoughtful comments. As the Reviewer’s suggestion, so we added some comment on the UCP-2 and UCP-3, the changes are as follows:

‘Furthermore, UCP-2 and UCP-3 are also involved in the regulation of energy metabolism as UCP homolues. In animals with a high-fat diet, exercise and hibernation, the mRNA expression levels of UCP-2 and UCP-3 increased accordingly to promote glucose and fatty acid metabolism, and the data from human genetic studies and measurements supports the contribution of UCP2 and UCP3 to resting energy expenditure. Therefore, enhancing the expression of UCPs may increase energy expenditure and provide a defensive effect on disordered lipid metabolism and damage.’

The change is on page 6.

3.This referee suggest to add a figure in paragraph 5 about the role of AMPK in browning of WAT.

Response: Thank you for your valuable and thoughtful comments. There have been many studies on the significant role of AMPK in the browning of white fat, which increases energy consumption and improves lipid metabolism. However, we are sorry that as for the study of the specific signaling mechanisms of pathways, our information is not enough to make a complete and comprehensive figure, and we will continue to study and improve in subsequent studies.

Other minor points

  1. Introduction, 12 lines TAG not TAH

Response: We are very sorry for our negligence on wrong abbreviations for TAG, and it has been corrected in the manuscript.

  1. Legend figure 1: add the full name for GPAT and PAP. Add diacylglycerol after the first DAG, not after the second.

Response: Thank you for your valuable and thoughtful comments. Even if we don't list them all, we've tried our best to revise our manuscript according to the comments. Here we did not list the changes but marked in red in revised paper.

  1. page 5, 5 lines from bottom; please replace to enjoy with to develope

Response: Thank you for your valuable and thoughtful comments, and it has been replaced in the manuscript.

  1. page 8, 15 lines: irisin not IRISIN

Response: Thank you for your valuable and thoughtful comments, and it has been replaced in the manuscript.

  1. page 8, 27 lines: add Ventromedial Nucleus of the Hypothalamus (VHM)

Response: We are very sorry for our negligence of unexplained abbreviations for the first time used, and it has caused unnecessary trouble to your reading. Here we did not list the changes of the total name of abbreviations but marked in red in revised paper.

Please see the attachment about the revised manscript.

We would like to express our great appreciation for your comments on our paper.

Thank you and best regards.

Yours sincerely,

Lei Zhang, Ph.D.
